# Modeling microbial metabolic trade-offs in a chemostat

Zhiyuan Li[1,2,3,4], Bo Liu[5], Sophia Hsin-Jung Li[6], Christopher G. King[7], Zemer Gitai[6], Ned S. Wingreen[6,8]*

1 Center for Quantitative Biology, Peking University, Beijing, China, 2 Peking-Tsinghua Center for Life Sciences, Peking University, Beijing, China, 3 Center for the Physics of Biological Function, Princeton University, Princeton, New Jersey, United States of America, 4 Princeton Center for Theoretical Science, Princeton University, Princeton, New Jersey, United States of America, 5 Yuanpei College, Peking University, Beijing, China, 6 Department of Molecular Biology, Princeton University, Princeton, New Jersey, United States of America, 7 Department of Physics, Princeton University, Princeton, New Jersey, United States of America, 8 Lewis-Sigler Institute for Integrative Genomics, Princeton University, Princeton, New Jersey, United States of America

* wingreen@princeton.edu

**Data Availability Statement:** All relevant data are within the manuscript and its Supporting Information files.A repository of all programs used to generate the results can be found at: https://github.com/zhiyuanli1987/Qbiotoolbox.git

## Abstract

Microbes face intense competition in the natural world, and so need to wisely allocate their resources to multiple functions, in particular to metabolism. Understanding competition among metabolic strategies that are subject to trade-offs is therefore crucial for deeper insight into the competition, cooperation, and community assembly of microorganisms. In this work, we evaluate competing metabolic strategies within an ecological context by considering not only how the environment influences cell growth, but also how microbes shape their chemical environment. Utilizing chemostat-based resource-competition models, we exhibit a set of intuitive and general procedures for assessing metabolic strategies. Using this framework, we are able to relate and unify multiple metabolic models, and to demonstrate how the fitness landscape of strategies becomes intrinsically dynamic due to species-environment feedback. Such dynamic fitness landscapes produce rich behaviors, and prove to be crucial for ecological and evolutionarily stable coexistence in all the models we examined.

## Author summary

Metabolism is the means by which organisms extract and process nutrients from the environment to live and grow. Various metabolic tasks cost cellular resources such as proteins and energy, while the total amount of resources within a cell are limited. In particular, microbial organisms–like companies with a tight budget–need to strategically allocate their limited resources. How can one assess the "effectiveness" of different microbial resource allocation strategies? Strategies are usually evaluated by their corresponding growth rates under a fixed environment, as if there were a static "fitness landscape". In our work, we demonstrate that even in a system as simple as a chemostat, assessing metabolic strategies is more complex than this static-landscape view: as microbes also shape their

**Funding:** This work was supported by the National Institutes of Health Grant R01GM082938 (NW, nih. org) and by the National Science Foundation, through the Center for the Physics of Biological Function (nsf.org, PHY-1734030).The funders had no role in study design, data collection and analysis, decision to publish, or preparation of the manuscript.

**Competing interests:** The authors have declared that no competing interests exist.

shared chemical environment, the fitness landscape becomes dynamic, and successful metabolic strategies are the ones that create a landscape that places themselves at the top. Focusing on how species create their own environment, we provide a geometric approach that unifies a variety of chemostat-type resource-competition models. Our approach yields an intuitive way of assessing the success of competing metabolic strategies, and shows how rich ecological dynamics can arise from species-environment feedback. Our work offers insights into the regulation and evolution of microbial metabolic strategies, and has implications for the biodiversity found in nature.

## Introduction

The way microbes respond to and shape their local environment influences their community structure [1]. Such microbe-environment interactions depend on the allocation strategies of cells, i.e., how a cell allocates its internal resources into various cellular functions, such as transport, assimilation, reproduction, motility, maintenance, etc. [2]. Within a microbial cell, energy and biomass are limited, and trade-offs always exist in allocating these valuable internal resources into the various functions required for cell growth. Therefore, the growth rate of cells cannot increase without bound. Rather, evolution acts on cells' internal resource allocation–primarily the production of proteins and nucleic acids–to optimize growth and survival [3, 4]. To this end, in response to environmental changes, microbes rapidly adjust their metabolic strategies. For example, the yeast *Saccharomyces cerevisiae* switches from fermentation to respiration upon glucose depletion [5], and *Escherichia coli* exhibits drastically different ribosome content between different nutrient conditions [6, 7]. Moreover, in laboratory long-term evolution studies of microbes, adaptive mutations consistently emerge that reshape metabolism [8–11]. Such short-term and long-term adjustments of metabolic strategies presumably confer fitness benefits, and it is important to map metabolic strategies onto these benefits to better understand the regulation and evolution of microbial metabolism.

To investigate the metabolic behavior of microorganisms, the convergence towards steady state makes the chemostat an ideal experimental system [12, 13]. In a chemostat, fresh nutrients are supplied at a constant rate, while medium with cells is removed at the same rate to maintain constant volume. The metabolite concentrations in the chemostat constitute the chemical environment directly perceived by cells, and determine their growth rates. Importantly, cells also shape this environment through their consumption and secretion of metabolites. One advantage of a chemostat is the automatic convergence of cellular growth rates towards the controlled dilution rate. This convergence occurs through negative feedback between microbes and their environment: the higher the population, the worse the chemical environment, and the slower the growth rate. As a result (provided the nutrient supply allows for faster-than-dilution growth to prevent "washout"), the cells in the chemostat will reach the steady-state population that sustains growth at the dilution rate [14, 15]. This stabilization of the cellular growth rate at the controlled dilution rate facilitates precise characterization of cellular physiology in a constant chemical environment. However, it also imposes challenges in quantitatively understanding the advantages and disadvantages of various metabolic strategies: if all metabolic strategies lead to identical growth rates in a chemostat, how should we evaluate whether one strategy is "better" or "worse" than another? If strategies can be compared, are there "best" strategies, and how do these optima shift as the experimental conditions such as nutrient-supply concentrations and dilution rates change?

In nature, including in chemostat-like ecosystems such as lakes and rivers, microbes must continually compete for survival [14]. If we evaluate metabolic strategies by the outcome of competition between species adopting these strategies, many insights can be gained from theoretical ecology. For example, resource-competition models have provided a simple context to explore competition dynamics in chemostat-like ecosystems. In such models, species interact only indirectly via consumption (and sometimes production) of a common pool of nutrients. A steady state can be reached if the species present can shape the nutrient concentration to support a growth rate equal to their dilution or death rate [16]. Resource-competition models underpin many ecosystem theories including contemporary niche theory as pioneered by MacArthur [17], popularized by Tilman [16, 18], and extended by Chase and Leibold [19]. A central component of contemporary niche theory is a graphical approach, generally consisting of three components: zero net growth isoclines (ZNGIs) in chemical space, an impact vector representing a species' nutrient consumption, and a supply point to describe the external resource supply [20]. This graphical approach is a powerful and intuitive way of assessing the outcome of competition, yet it is not yet commonly utilized in understanding microbial metabolic strategies with trade-offs.

Resource-competition models focusing on various aspects of cellular metabolism vary in their assumptions regarding species-environment interactions, and can lead to diverse results for community structure and population dynamics. In a model where species compete for essential resources, different nutrient requirements can produce intrinsically oscillatory or even chaotic dynamics [21, 22]. Alternatively, cross-feeding [23–25] can promote coexistence, while preferential nutrient utilization [26] can lead to multistability. With metabolic trade-offs, a model in which growth rate is additive in imported nutrients self-organizes to a state of unlimited stable coexistence [27], while another model with convertible essential nutrients also allows evolutionarily stable coexistence but with a limited number of species [28]. This large variety of models and the richness of possible behaviors raises the question of unification: is there a simple framework that consolidates this diverse group of models into one easily understandable picture?

Continuity of the strategy space adds another layer of complexity in characterizing the "best" metabolic strategy or strategies. With infinite possibilities for allocating cellular resources, how should one pinpoint the optimal ones? Adaptive dynamics in ecological theory, also known as evolutionary invasion analysis, provides valuable guidance [29]. This mathematical framework addresses the long-term evolution of traits in asexually reproducing populations by quantifying the fitness of each "trait" as a function of population composition. In this framework, "invasion fitness" is defined as the net-growth rate of a new variant when it is introduced into the indigenous population in an infinitesimally small amount, and a population allowing only non-positive invasion fitness for any new variant is considered to be "evolutionarily stable" [30–32]. Such an evolutionarily stable point is valuable for defining optimal strategies, as a community adopting the most suitable metabolic strategies should not be invasible by any other strategies. As microbes in nature frequently experience environmental heterogeneity, it is important to understand whether and how such "optimal metabolic strategies" change with external conditions. Nevertheless, in the standard modeling framework of adaptive dynamics, the environment is implicit, and species directly act on each other without the realistic constraints imposed by competition for resources. Combining the concept of invasion fitness with explicit competition for resources, under the assumption of metabolic trade-offs, could therefore bring new insights into microbial metabolic strategies and community assemblies.

In this work, we present a mathematical framework for analyzing competition for resources among various metabolic strategies in a chemostat setting. We combine and extend the

graphical tools from resource-competition theory and the invasion-fitness approach from adaptive dynamics, to relate and unify multiple models for microbial metabolic trade-offs. This combination provides an intuitive scheme to evaluate strategies under various external conditions. The center of the framework is the role of species in creating their own environment. Firstly, the chemical environment shaped by an indigenous species through growth and consumption can be inviting or prohibiting to an invader species, depending on the geometric relationship between the "zero net growth surface" of the invader and the environment created by the indigenous species. This geometry-dependent fitness leads to a general criterion for whether an invader species can establish itself in the steady-state environment created by the indigenous species, which we call the "rule of invasion". In evaluating a continuum of strategies, the relationship between each strategy and its instantaneous growth rates defines a "fitness landscape", whose shape depends on the chemical environment. The deformability of this fitness landscape, i.e. its dependence on which species are present, can prevent single strategies from unconditional dominance [33]. We demonstrate how such deformable fitness landscapes can lead to rich ecosystem dynamics, including mutual invasion, multistability, and oscillations, and how all of these behaviors can be simply related via a geometric representation. Moreover, from the environment-dependent fitness landscape, we can define non-invasible/optimal metabolic strategies–namely, one or more strategies that construct a fitness landscape that places themselves on the top. The mathematical framework we present establishes an intuitive mapping from various metabolic models to the dynamical fitness landscape and population dynamics. Additionally, it reveals long-term implications, particularly in clarifying the general conditions for coexistence on both ecological and evolutionary time scales.

## Results

### Metabolic trade-offs and metabolic strategies

As discussed above, microorganisms need to allocate their limited internal resources into different cellular functions. In our models, we use $a_j$ to denote the fraction of internal resources allocated to the $j$-th metabolic function, with $\overrightarrow{\alpha} = (\alpha_1, \alpha_2 \ldots)$ representing a metabolic strategy. As a simple representation of the limited internal resources, an exact metabolic trade-off is assumed, such that $\sum_j \alpha_j = 1$. All possible values of $\overrightarrow{\alpha}$ define a continuous spectrum of metabolic strategies, which we name the strategy space. One major goal of our work is to construct a general and intuitive framework for evaluating strategies for a broad range of different metabolic models and experimental conditions.

### Geometrical representation of how strategies interact with the environment

One way to evaluate metabolic strategies is by comparing the competitiveness of "species" with fixed strategies in chemostat-type resource-competition models. In an idealized model of a chemostat (Fig 1A), $p$ types of nutrients are supplied at rate $d$ and concentrations $\overrightarrow{c}_{\text{supply}} = (c_{1,\text{ supply}}, c_{2,\text{ supply}}, \ldots c_{p,\text{ supply}})$, meanwhile cells and medium are diluted at the same rate $d$. The chosen values of $d$ and $\overrightarrow{c}_{\text{supply}}$ constitute the "external condition" for a chemostat. Accordingly, the biomass density $m_\sigma$ of species $\sigma$ adopting strategy $\overrightarrow{\alpha}_\sigma$ in the chemostat obeys:

$$\frac{dm_\sigma}{dt} = m_\sigma \cdot \left( g(\overrightarrow{c}, \overrightarrow{\alpha}_\sigma) - d \right). \tag{1}$$

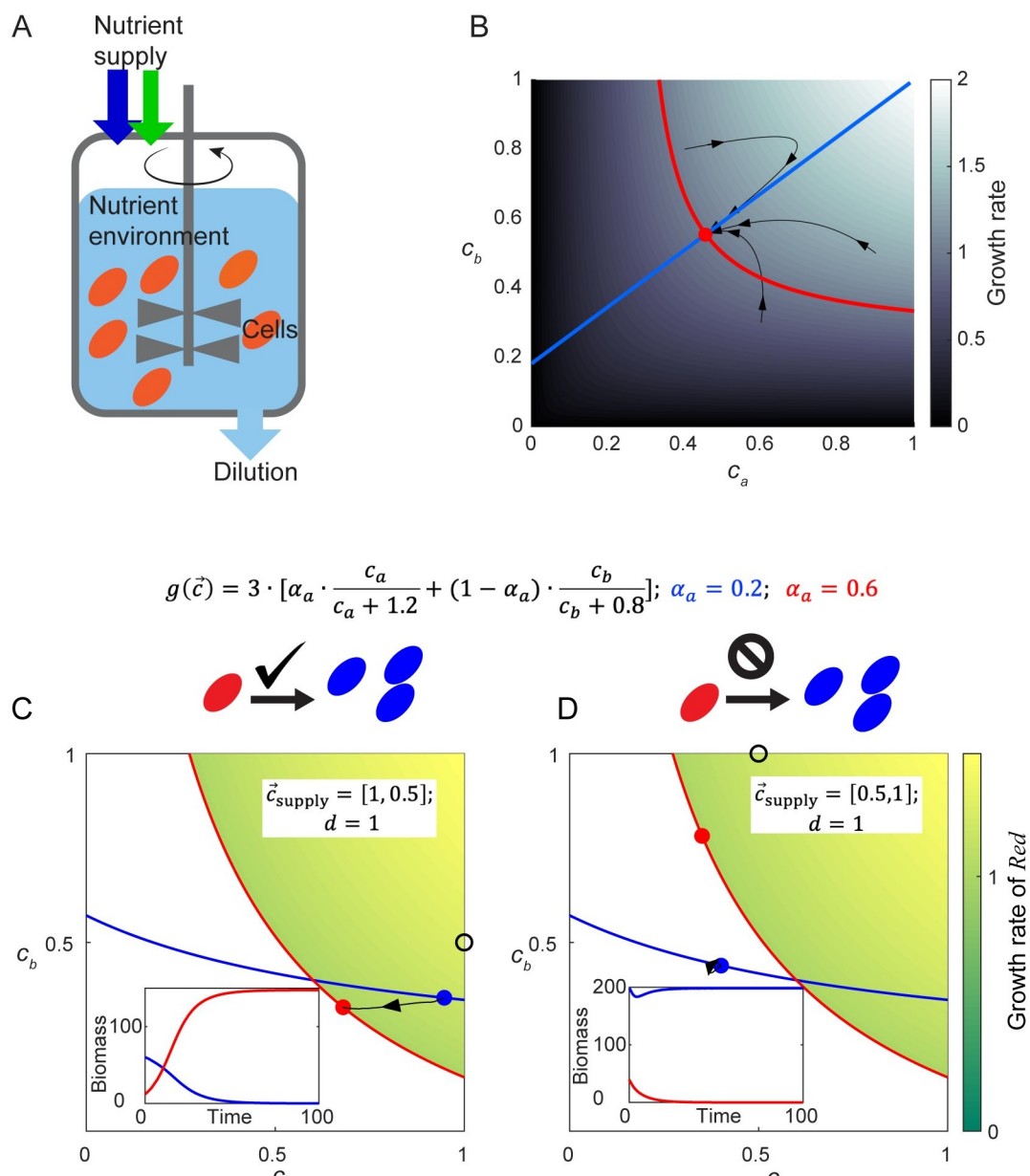

**Fig 1. Chemostat behavior represented in chemical space.** A. Schematic diagram of a chemostat occupied by a single microbial species. In the well-mixed medium (pale blue) of a chemostat, cells (orange ellipses) consume nutrients and grow. An influx of nutrients with fixed concentrations (blue and green arrows) is supplied at the same rate as dilution, keeping the medium volume constant. B. Visual representation of how a species creates its own chemostat environment. Background color indicates the growth rate of cells as a function of metabolite concentrations $c_a$ and $c_b$, with the growth contour shown by the red curve. The flux-balance curve is shown in blue. Black curves with arrows show the time trajectories of chemostat simulation. C. Example of successful invasion of the indigenous species *Blue* by the invader species *Red*. A small amount of species *Red* is introduced to a steady-state chemostat of species *Blue*. Growth contours and steady-state environments of species *Blue* and species *Red* are shown as curves and dots in the corresponding colors (colored background indicates the "invasion zone" of *Red*, and represents the growth rate of *Red* in this zone). The supply condition is marked by a black circle. Black curves with arrows show the time trajectory of the invasion in chemical space. Inset: time course of species biomass in the chemostat during the invasion. D. Same as (C), except that because the supply condition (black circle) is different, the attempted invasion by species *Red* is unsuccessful.

The concentration $c_i$ of the $i$-th nutrient is a variable, influenced by its rate of consumption $I_i$ per cell volume. In a chemostat occupied by a single species $\sigma$, the changing value of $c_i$ satisfies

$$\frac{dc_i}{dt} = d \cdot \left( c_{i,\ \text{supply}} - c_i \right) - m_\sigma / r \cdot I_i\left( \overrightarrow{c}, \overrightarrow{\alpha}_\sigma \right), \tag{2}$$

where $r$ is a constant representing the biomass per cell volume (See S1 Appendix for details). A negative value of $I_i$ corresponds to secretion of the metabolite from cells into the environment.

In this manuscript, we define $\overrightarrow{c}$ as the "chemical environment", and all possible values of $\overrightarrow{c}$ constitute the "chemical space". Eqs (1) and (2) represent a general chemostat model with a single species. The simplicity of the chemostat has inspired many theoretical studies of resource competition. Different model assumptions about how species grow (Eq (1)) and consume nutrients (Eq (2)) have produced a variety of intriguing behaviors and conclusions. However, the origins of these differences are not always simple to discern. To facilitate the evaluation of metabolic strategies, we next present a geometric representation that allows ready visualization of the feedback between species and the environment in a chemostat. Intuitively, the steady state created by a single species can be visualized by the intersection of two nullclines, derived from Eq (1) and Eq (2), respectively (details in S1 Appendix):

First, setting Eq (1) to zero leads to a $p$-dimensional version of the ZNGI, which we name the "*growth contour*". For a given metabolic strategy $\overrightarrow{\alpha}_\sigma$, the growth-rate function $g(\overrightarrow{c}, \overrightarrow{\alpha}_\sigma)$ maps different environments in the chemical space onto different growth rates (background color in Fig 1B). At steady state, the relation $dm_\sigma/dt = 0$ (Eq (1)) requires the growth rate to be exactly equal to the dilution rate (assuming nonzero cell density). Therefore, the contour in chemical space satisfying $g(\overrightarrow{c}, \overrightarrow{\alpha}_\sigma) = d$ indicates all possible environments that could support a steady state of the strategy $\overrightarrow{\alpha}_\sigma$ (red curve in Fig 1B, and orange, red, deep red curves in S1B Fig). This contour reflects how the chemical environment determines cell growth.

Secondly, the nullcline derived from Eq (2) reflects the impact of cellular metabolism on the chemical environment. At steady state, the nutrient influx should be equal to the summation of dilution and cellular consumption. When Eq (2) is set to zero, varying values of cell density $m$ lead to different values of $\overrightarrow{c}$ (Eq. (S5)) constituting a one-dimensional "flux-balance curve" in chemical space (blue curve in Fig 1B, and purple, cyan, and blue curves in S1A Fig).

At the intersection of the growth contour and the flux-balance curve, the steady-state chemical environment $\overrightarrow{c}_{\sigma,\text{ss}}$ is created by the species $\sigma$ (Fig 1B, red dot). Changes in conditions $d$ and $\overrightarrow{c}_{\text{supply}}$ influence the shapes of the growth contour and the flux-balance curve separately, enabling clear interpretation of chemostat experiments under varied conditions (S2 Fig, details in S1 Appendix).

This graphical approach provides an intuitive understanding of how chemostat experiments can be controlled and interpreted (See S1 Appendix for details). More importantly, it enables an intuitive picture for the outcomes of competitions among multiple species.

## Evaluating strategies by the rule of invasion, and the environment-dependent fitness landscape

We use the outcome of competition between species to evaluate metabolic strategies, assuming each species $\sigma$ adopts a fixed strategy $\overrightarrow{\alpha}_\sigma$. We first focus on the outcome of invasion. Similar to several previous works that use invasion to assess the stability of a consortia [29, 34], here "invasion" is defined as the introduction of a very small number of an "invader" species to a steady-state chemostat already occupied by a set of "indigenous" species. The invasion growth

rate is quantified by the instantaneous growth rate of the invader species at the moment of introduction. Unlike adaptive dynamics, the invasion growth rate is evaluated with respect to the chemical environment created by the indigenous species, rather than with respect to the population composition of the indigenous species [30–32].

In the chemical space, the outcome of an invasion can be summed up by a simple geometric rule, as demonstrated in Fig 1C and 1D. The growth contour of the invader (species *Red*) separates the chemical space into two regions: an "invasion zone" where the invader grows faster than dilution (green-colored region in Fig 1C and 1D), and "no-invasion zone" where the invader has a growth rate lower than dilution. If the steady-state environment constructed by the indigenous species (species *Blue*) is located within the invasion zone of the invader, the invader will initially grow faster than dilution. Therefore, the invader will expand its population and the invasion will be successful (Fig 1C). By contrast, if the steady-state chemical environment created by the indigenous species lies outside of the invasion zone, the invasion will be unsuccessful (Fig 1D, same species as in Fig 1C but with a different supply condition, and therefore a different steady state). (See S1 Appendix for details.)

In the steady-state environment created by the indigenous species, each strategy $\vec{\alpha}$ has an invasion growth rate. We define an environment-dependent "fitness landscape" as the relation between the invasion growth rate and the metabolic strategy of invaders (Eqs. (S8)-(S9), see S1 Appendix for details). Different indigenous species can create different chemical environments, and thus give rise to different shapes of the fitness landscape.

## Mutual invasion, a flat fitness landscape, and unlimited coexistence

The rule of invasion allows for easy assessment of competition dynamics. The emergence of complex dynamics generally requires that competitiveness be non-transitive [33]. For example, if species *Red* can invade species *Blue*, that does not mean *Blue* cannot invade *Red*. Such mutual invasibility can be observed in substitutable-resource metabolic models, with a simple version illustrated in Fig 2A: two substitutable nutrients *a* and *b*, such as glucose and galactose, contribute linearly to biomass increase. Since a substantial investment of protein and energy is required for nutrient uptake, the model assumes an exact trade-off between the allocation of internal resources to import either nutrient. Specifically, a fraction of resources $\alpha_a$ is allocated to import *a* and a fraction $(1-\alpha_a)$ to import *b*. As shown in Fig 2B, while the steady-state environment created by *Blue* is located within the invasion zone of *Red*, the steady-state environment created by *Red* is also located within the invasion zone of *Blue*. According to the rule of invasion, each species can therefore invade the steady-state environment created by the other. In the face of such successful invasions, the only possible stable chemical environment for this system is at the intersection of two growth contours, where the two species can coexist.

The environment-dependent fitness landscape readily explains this coexistence: In this model, the fitness landscape is linear respect to $\alpha_a$, with the slope positively correlated with the difference between the steady state value $c_a^*$ and $c_b^*$ (See S1 Appendix for details). In the steady-state environment created by *Red* ($\alpha_a = 0.6$), *a* becomes scarce relative to *b*, and strategies with smaller $\alpha_a$ have higher fitness (Fig 2C, upper panel). In the steady-state environment created by *Blue* ($\alpha_a = 0.2$), the slope of the fitness landscape changes its sign, and strategies with larger $\alpha_a$ have higher fitness (Fig 2C, middle panel). Therefore, each species creates an environment that is more suitable for its competitor, which leads to coexistence.

Typically in resource-competition models, the number of coexisting species cannot exceed the number of resources [35–38]. This conclusion can be understood intuitively from the geometric approach: The steady-state growth of a species can only be achieved on the growth contour of this species. The stable-coexistence of *N* species can thus only occur at the intersection

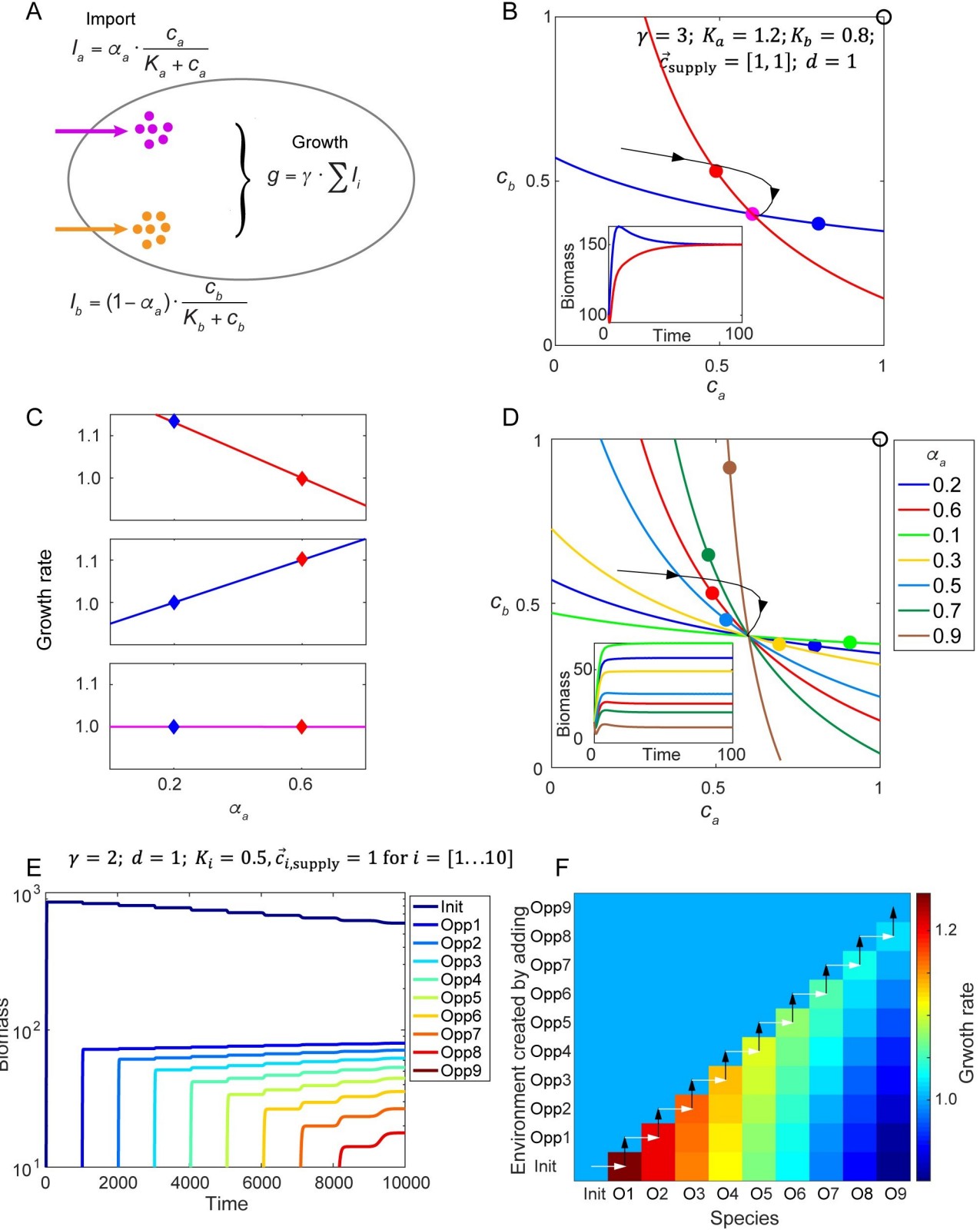

**Fig 2. Metabolic models with substitutable nutrients can achieve a flat fitness landscape.** A. Example of a metabolic model with a trade-off in allocation of internal resources for import of two substitutable nutrients, with both nutrients contributing additively to growth. Species *Red* and species

*Blue* allocate resources differently (indicated by parameter $\alpha_a$, see S1 Appendix). B. Growth contours and the steady-state environments created by *Red* or *Blue* alone, under the supply condition shown by the black circle. Black curve with arrows shows a trajectory in chemical space. Purple dot indicates the steady-state environment created by *Red* and *Blue* together. Lower inset: time course of species biomass. C. From upper panel to bottom panel: the fitness landscape created by *Red* alone (for the red dot in (B)), created by *Blue* alone (for the blue dot in (B)), and created by both species (for the purple dot in (B)). Diamonds mark the locations of *Red* and *Blue* strategies and their corresponding fitness in each fitness landscape. D. Growth contours and the species-specific steady-state environments for seven different species alone, under the supply condition shown by the black circle. Black curve with arrows shows a trajectory in chemical space. Lower inset: time course of species biomass in the chemostat. E. Population dynamics in a 10-dimensional chemical space. The chemostat is initially occupied by a species (Init) that has an arbitrarily assigned enzyme allocation strategy. Then, in the steady-state environment created by Init, the "opportunist" species (Opp 1) with the maximal growth rate in that environment is added to the chemostat. Subsequently, further opportunist species (Opp 2–9) are added sequentially to the steady states created by the existing consortia of species, until there is no further opportunist strategy with a growth rate higher than the dilution rate. F. The instantaneous growth rates of the 10 strategies from (E) under the steady-state environments created after adding each species to the existing consortium. Each white arrow indicates the addition of the new species with the fastest growth rate in that environment; black arrows indicate the change of the steady-state chemical environment caused by adding these new species.

of their $N$ corresponding growth contours. Generally, an $N$-dimensional chemical space allows a unique intersection of no more than $N$ surfaces, and the diversity of species is therefore bounded by the number of metabolites in the environment. This theoretical restriction on biodiversity, made formal as the "competitive exclusion principle", contradicts the tremendous biodiversity manifested in the real world [25, 39, 40]. There have been a multitude of theoretical efforts to reconcile this contradiction [21, 22, 24, 25, 28, 41].

Under the very simplified assumption of exact trade-offs, a special property of this metabolic model is that all growth contours intersect at a common point (Fig 2D). In the environment co-created by *Blue* and *Red* (Fig 2B, purple dot), which is the common intersection point for all growth contours, the fitness landscape becomes flat (Fig 2C, bottom panel). Therefore, in this system, once any pair of species with a mutual-invasion relationship constructs the steady-state chemical environment together, all species become effectively neutral. Subsequent works showed that with spatial structure [42], even non-exact trade-offs can lead to high species abundance by partially "leveling the playing field" among different metabolic strategies, showing the potential of a nearly-flat fitness landscape to promote biodiversity.

In order to understand whether a microbial community will evolve towards or away from a flat fitness landscape, we extended the metabolic model to incorporate 10 substitutable nutrients. We started the chemostat with a species with an arbitrarily assigned strategy (Init), and let it come to steady state. Then, in the steady-state environment created by species Init, the "opportunist" species (Opp 1) with the maximal growth rate in that environment was added to the chemostat. Subsequently, in the steady states created by the existing consortia, we identified the fastest growing opportunist species (Opp 2–9) and added them sequentially to the chemostat, until there was no further opportunist strategy with a growth rate higher than the dilution rate (Fig 2E). During this process, the newly selected opportunist can always invade without replacing any of the existing species (Fig 2E). It is analytically provable that these opportunist species exhibit all-or-none resource allocation strategies ($\alpha_i = \{0,1\}$) to maximize their growth rates (S3A Fig, S1 Appendix), and act as "keystone" species, similar to those defined in the work of Posfai et al., that expand the convex hull for coexistence [27]. As the opportunists specializing in different nutrients were selected and added to the chemostat one by one, more species start to acquire equal-to-dilution growth rates (Fig 2F), and the fitness landscape becomes more and more flattened (S3B Fig). Finally, this process of "evolution" of the community self-organizes towards multiple keystone species that completely flatten the fitness landscape and thus ensure unlimited coexistence.

This substitutable-resource metabolic model also suggests that non-stationary fitness landscape is the prerequisite for interesting ecological dynamics. Another example comes from a slightly modified metabolic model, where enzymes are also required to convert the imported

raw materials into biomass, which can form a "rock-paper-scissors"-type invasion loop (See S1 Appendix and S4 Fig). This loop leads to oscillatory population dynamics with an ever-changing fitness landscape, similar to the oscillatory dynamics demonstrated by Huisman et al. [21].

## Multistability and the chain of invasion

When species create environments that are more favorable for their competitors, mutual-invasion can occur. Can species create environments that are hostile to their competitors, and if so what will be the consequences?

Fig 3A shows a simple metabolic model with two essential nutrients *a* and *b*, such as nitrogen and phosphorus (see S1 Appendix for details). Similar to the model in Fig 2A, the model assumes a trade-off between the allocation of internal resources to import nutrients, so that a resource allocation strategy is fully characterized by the fraction of resources $\alpha_a$ allocated to import nutrient *a*. The growth rate is taken to be the minimum of the two input rates [43]. As shown in Fig 3B, two species, *Red* and *Blue*, each creates a chemical environment outside of the invasion zone of each other. According to the rule of invasion, neither can be invaded by the other. Therefore, the steady state of the community depends on initial conditions–whichever species occupies the chemostat first will dominate indefinitely. It is worth noting that despite the fact that coexistence is excluded in a single chemostat under this metabolic model, the ability of species to create a self-favoring environment allows the spontaneous emergence of spatial heterogeneity and coexistence in an extended system with multiple linked chemostats (S6 Fig). In ecology, the spontaneous emergence of spatial heterogeneity has been shown for species with the capacity to construct their own niches [37, 44], and the chain of chemostats provides a simple model for such spatial coexistence.

The nonmonotonic fitness landscape can produce a chain of invasion. From the perspective of the strategy-growth relationship (Fig 3B, inset), species *Red* ($\alpha_a = 0.65$) creates a fitness landscape where small $\alpha_a$ is disfavored. Symmetrically, species *Blue* ($\alpha_a = 0.35$) creates a fitness landscape where large $\alpha_a$ is disfavored. However, neither *Red* nor *Blue* sits on the top of the fitness landscape each one creates (Fig 3C). In the fitness landscape created by *Blue*, a slightly larger $\alpha_a$ (green diamond in Fig 3C) has the highest growth rate. Consequently, species adopting the *Green* strategy can invade *Blue*. Nevertheless, species *Green* is not on the top of its own fitness landscape as an even larger $\alpha_a$ (yellow diamond in Fig 3C) maximizes the growth rate in the environment created by *Green*. If we repeatedly perform the process of adding the fastest-growing species to the chemostat, as in the previous section, the newly added species always outcompetes and replaces the former species. A series of replacements by the fastest-growing species in the environment created by the former species creates a chain of invasion (Fig 3E), which finally leads to a balanced enzyme budget with $\alpha_a = 0.5$. Yet, even for this final strategy that cannot be replaced by any other strategies ($\alpha_a = 0.5$), it cannot invade strategies just three steps earlier (Fig 3E).

In this particular model after four steps of replacement, multistability appears. The species with $\alpha_a$ marked by *Deep Purple*, which is reached by the chain of invasion going from *Blue* to *Green* to *Yellow* to *Deep Green* cannot invade the original species *Blue* (Fig 3C). A similar relationship holds between *Cyan* and *Red*. Actually, any set of species in Fig 3E that are not directly linked by the arrow of invasion will exhibit multistability. This phenomenon highlights the difference between ecological stability and evolutionary stability: Ecologically, as both *Blue* and *Deep Purple* create a fitness landscape where the other species grows slower than dilution, they constitute a bistable system. However, evolutionarily, "mutants" with slightly larger $\alpha_a$ can invade *Blue*, eventually driving the system towards *Deep Purple*.

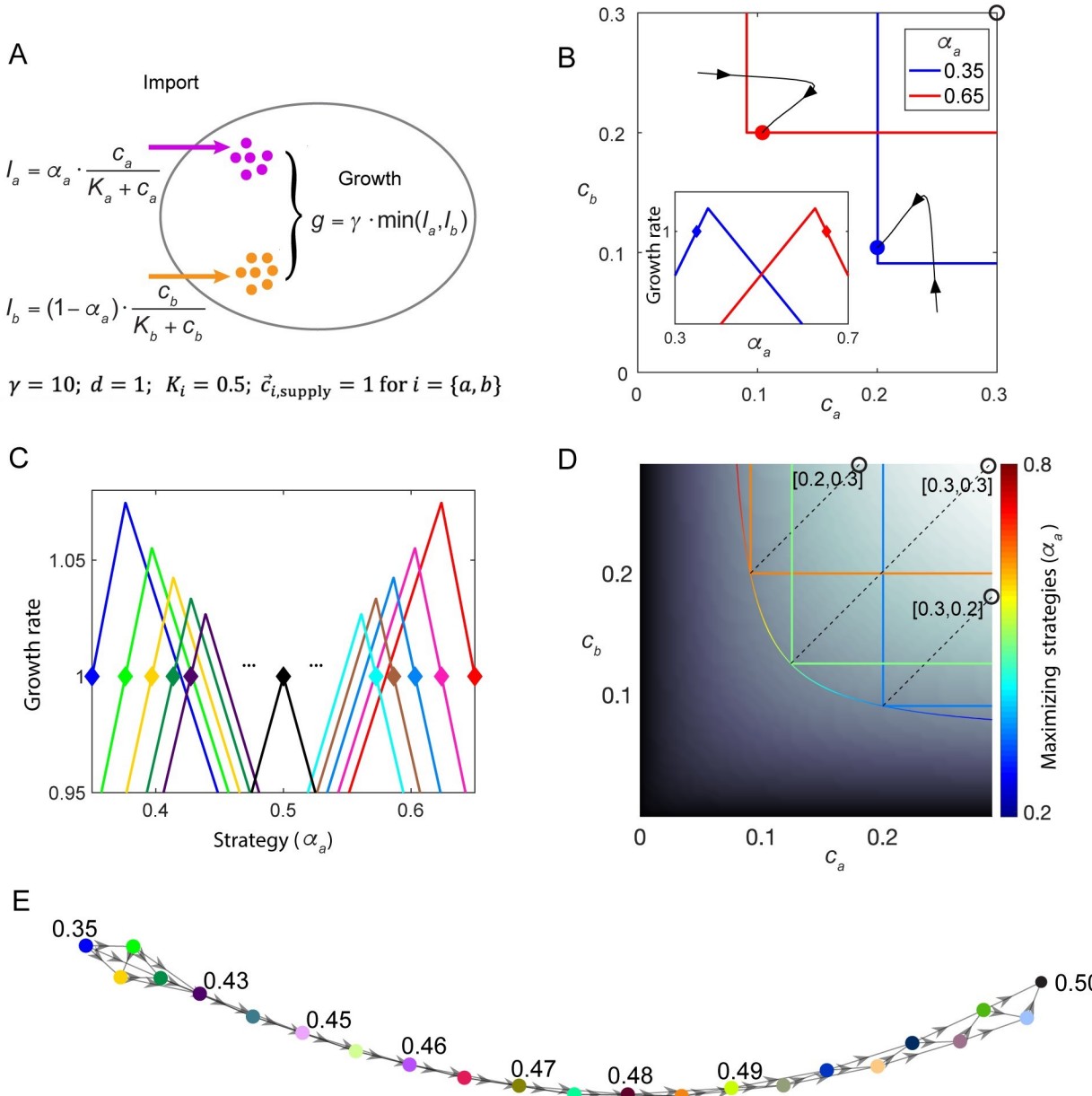

**Fig 3. Multistability, chain of invasion, and non-invasible strategy.** A. Example of a metabolic model with a trade-off in allocation of resources for import of two essential nutrients, with the lower of the two import rates determining growth rate. Species *Red* and species *Blue* allocate resources differently (indicated by parameter $\alpha_a$, see S1 Appendix). B. Bistability of the system in (A) shown in chemical space. Black curves with arrows show the trajectories of simulations with different initial conditions. Inset: the fitness landscape created by species *Red* or *Blue* alone, with colors corresponding to the steady-state environments shown by colored dots in the main panel. C. The evolving fitness landscape. Fitness landscape created by species with different internal resource allocation strategies (marked by diamond shapes). Starting from species *Blue*, the species having the highest growth rate in the steady-state fitness landscape created by the "former" species is selected. This creates a chain of invasion from *Blue* to *Light Green*, *Yellow*, *Deep Green*, *Deep Purple*, all the way (intermediate processes omitted) to the species *Black*, which places itself on the peak of its own fitness landscape. The same procedure is also performed starting with species *Red*. D. Depiction of non-invasible strategies under different supply conditions. Black-white background indicates the maximal growth rate of the model in (A) under each environment, and the contour of maximal growth rates contains different strategies (represented by red-to-blue color). Growth contours of three species adopting one of the "maximizing strategies" are colored by their strategies. The supply conditions allowing these strategies to be "non-invasible" (supply lines) are marked by dashed black lines. E. Chain of invasion. Addition of the strategy with the fastest growth rate under the steady-state environment created by the existing consortium, as indicated in (C), is repeated 22 times. The sequentially added strategies are marked by colored circles, with the value of $\alpha_a$ give for some representative strategies. An arrow from node *j* to node *i* indicates that strategy *j* can invade the environment created by strategy *i*.

## Non-invasible strategies

In this model, with symmetric parameters, the only evolutionarily stable strategy is $\alpha_a = 0.5$ (black diamond in Fig 3C). This is the only strategy that locates itself on the top of the fitness landscape it creates, and therefore cannot be invaded by any other species. This simple model demonstrates a general definition of optimal (aka evolutionarily stable or non-invasible) strategies: those strategies that create a fitness landscape which places themselves on the top (Eq. (S10)).

A chemical environment defines a fitness landscape, and the steady-state chemical environment created by the species present is influenced by supply condition, dilution rate, and the details of cell metabolism. Therefore, different chemostat parameters and different metabolic models lead to different optimal strategies. In the following, we described a generally applicable protocol for obtaining the non-invasible strategies, using the metabolic model in Fig 3A as the example (Fig 3D, details in S1 Appendix):

First, under a chemical environment $\vec{c}$, the maximal growth rate $g_{max}(\vec{c})$ (background color in Fig 3D) and the corresponding resource allocation strategy $\vec{\alpha}_{max}(\vec{c})$, defined as a "maximizing strategy", can be obtained analytically or via numerical search through the strategy space (Eq. (S11)). $g_{max}(\vec{c})$ and $\vec{\alpha}_{max}(\vec{c})$ are independent of the chemostat parameters $\vec{c}_{supply}$ and $d$.

Second, the "maximal growth contour" for dilution rate $d$ is defined as all chemical environments $\vec{c}$ that support a maximal growth rate of $d$ (Eq. (S12)). Different maximizing strategies $\vec{\alpha}_{max}(\vec{c})$ exist at different points of the maximal growth contour, as shown by the colors of the curve in Fig 3D. In a multi-species ecosystem, possible steady states can only occur at the outermost surface of the multiple species' growth contours, as highlighted in S3C Fig. By definition, the maximal growth contour is the outermost surface that envelops all growth contours, so that chemical environments on the maximal growth contour are outside of the invasion zone of any strategy. Therefore, if a species is able to create a steady-state environment on the maximal growth contour, it cannot be invaded.

Finally, different $\vec{c}_{supply}$ form different maximal flux-balance curves (Eq. (14)), which intersect with the maximal growth contour at one point $\vec{c}_{opt}$. Species $\vec{\alpha}_{max}(\vec{c}_{opt})$ that adopt the maximizing strategy at $\vec{c}_{opt}$ create the environment $\vec{c}_{opt}$, and are therefore immune to invasion. Under different $\vec{c}_{supply}$, different species become non-invasible (e.g., orange, green, and blue growth contours in Fig 3D).

## Conditions for evolutionarily stable coexistence

Given $d$ and $\vec{c}_{supply}$, the maximal growth contour and the maximal flux-balance curve are unique, therefore there is only one $\vec{c}_{opt}$. Does the uniqueness of $\vec{c}_{opt}$ imply a single evolutionarily stable species? Or is coexistence still possible even in the face of evolution? In a recent work [28], this question was addressed by modeling a population of microbes competing for steadily supplied resources. Through *in-silico* evolution and network analysis, the authors found that multiple species with distinct metabolic strategies can coexist as evolutionarily-stable co-optimal consortia, which no other species can invade.

Using a simplified version of Taillefumier et al.'s model (Fig 4A), we employ the graphical approach to help identify the requirements for such evolutionarily-stable coexistence and the role of each species in supporting the consortium. In this model, at the cost of producing the necessary enzymes, cells are not only able to import external nutrients, but can also convert any one of the internal nutrients into any other. Meanwhile, nutrients passively diffuse in and

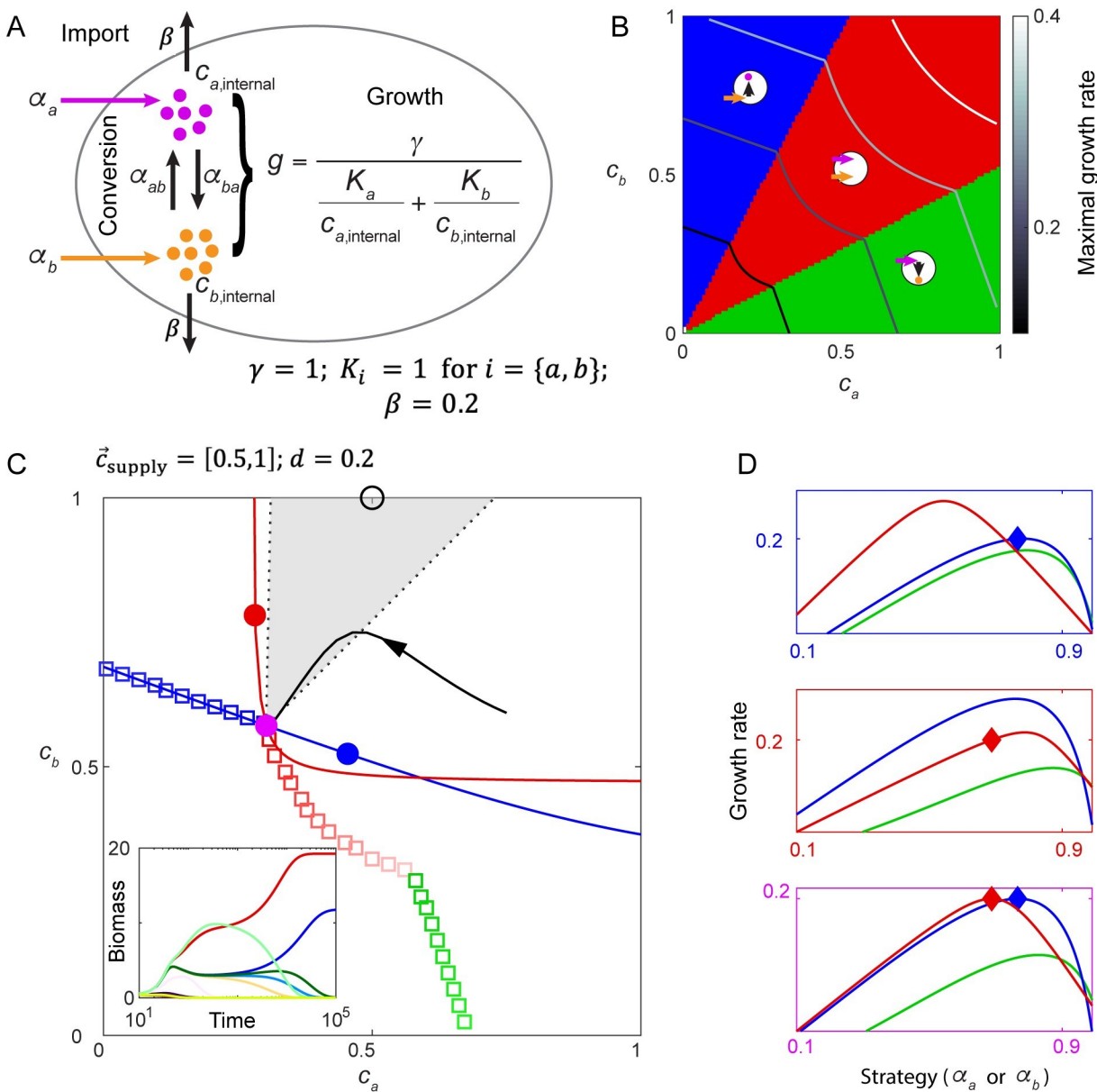

**Fig 4. Non-invasible cartels.** A. Metabolic model with a trade-off in allocation of internal resources for import of two nutrients plus their interconversion, with both nutrients necessary for growth. B. Three subclasses of maximizing metabolic strategies in chemical space are indicated by background color, and circles with arrows illustrate the metabolic strategies of each subclass. The maximal growth contours for four growth rates (0.1, 0.2, 0.3, 0.4) are marked by gray colors. C. Two maximizing strategies co-creating a non-invasible steady state. At dilution rate $d = 0.2$, the maximal growth contour and the corresponding maximizing strategies are shown as colored squares. At a discontinuous point of the growth contour, the supply lines of two distinct metabolic strategies (*Red* and *Blue*) span a gray region, where any supply condition (e.g. black circle) requires the two maximizing strategies to co-create the environment on the discontinuous point. Red and blue dots mark the environments created by species *Red* and species *Blue* alone, and the purple dot marks the environment co-created by *Red* and *Blue*. Black curve with arrows shows a trajectory in chemical space. Inset: competition dynamics of species *Red* and species *Blue* together with 10 other maximizing species with different strategies. D. The fitness landscapes for the three environments in (C) indicated by corresponding box colors. For class Green and Red, the strategy is represented by $\alpha_a$, for class Blue, the strategy is represented by $\alpha_b$.

out of the cell. The internal concentrations of nutrient *a* and nutrient *b* are both essential for cell growth (see S1 Appendix for detail). Therefore, metabolic trade-offs in this system have four elements: the fraction of internal resources allocated to import nutrient *a* ($\alpha_a$) or nutrient

$b$ ($\alpha_b$) and/or convert one nutrient into another ($\alpha_{ab}$ converts internal $b$ into $a$, and $\alpha_{ba}$ converts internal $a$ into $b$). Each species is defined by its internal resource allocation strategy $\vec{\alpha} = (\alpha_a, \alpha_b, \alpha_{ab}, \alpha_{ba})$.

Following the general protocol described in the previous section, we first identified the maximal growth rates $g_{max}(\vec{c})$ and the corresponding strategy or strategies $\vec{\alpha}_{max}(\vec{c})$ at each point $\vec{c}$ in the chemical space, and generated maximal growth contours for different dilution rates (Fig 4B). The maximal growth contours are not smoothly continuous, nor are the corresponding strategies. In chemical space, three distinct sectors of maximizing strategies appear (Fig 4B, S7A Fig): When nutrient $a$ is very low compared to $b$, the maximizing strategy is a "*b-a* converter" which imports $b$ and converts it into $a$ (blue sector, only $\alpha_b$ and $\alpha_{ab}$ are non-zero). Symmetrically, when $a$ is comparatively high, the optimal strategy is a "*a-b* converter" (green sector, only $\alpha_a$ and $\alpha_{ba}$ are non-zero). Otherwise, the maximizing strategy is an "importer" which imports both nutrients without conversion (red sector, only $\alpha_a$ and $\alpha_b$ are non-zero). On the border between sectors, the maximal growth contour has a discontinuous slope.

Optimal coexistence occurs at these discontinuous points. If an environment point $\vec{c}_0$ is located in a continuous region of the maximal growth contour, only one maximizing strategy $\vec{\alpha}_{max}(\vec{c}_0)$ exists for that environment (maximizing strategies along the maximal growth contour are indicated by colored squares in Fig 4C). Supply conditions that make $\vec{\alpha}_{max}(\vec{c}_0)$ the optimal strategy (i.e. allow $\vec{\alpha}_{max}(\vec{c}_0)$ to create the steady-state environment $\vec{c}_0$) constitute the supply line for $\vec{c}_0$ and $\vec{\alpha}_{max}$.

The optimal coexistence of species *Blue* and species *Red* can be understood intuitively from the dynamic fitness landscape. Given a chemical environment, the relation between $\alpha_a$ and the growth rate of an importer (red curve) or an *a-b* converter (green curve), and that between $\alpha_b$ and the growth rate of a *b-a* converter (green curve) constitute the fitness landscape of species adopting different sectors of maximizing strategies (Fig 4D). In the environment created by species *Blue* (blue dot in Fig 4C), not only will some importers grow faster than *Blue*, species *Blue* (strategy marked by blue diamond) is not even on the fitness peak of its own class (Fig 4D, upper panel). Similarly, in the environment created by species *Red*, the strategy of *Red* is not at the top of the fitness landscape (Fig 4D, middle panel). By contrast, in the environment co-created by species *Blue* and *Red* (purple dot in Fig 4C), their strategies are at the top of the fitness landscapes of their own classes and at equal height. For all supply conditions in the gray region, species *Blue* and species *Red* jointly drive the nutrient concentrations to the discontinuous point of the optimal growth contour, and thereby achieve evolutionarily stable coexistence.

## Species creating a new nutrient dimension, and evolutionary stability with or without cross-feeding

One possible solution to the competitive-exclusion paradox is the creation of new nutrient "dimensions" by species secreting metabolites that can be utilized by other species. For example, *E. coli* secretes acetate as a by-product of glucose metabolism. Accumulation of acetate impedes the growth of *E. coli* on glucose [45], but the acetate can be utilized as a carbon source, e.g. by mutant strains that emerge in long-term evolution experiments [46, 47]. Recently, several modeling works investigated community structures when microbes create metabolic niches by secreting metabolic byproducts, and found that cross-feeding is capable of supporting high ecological diversity [23, 48]. Nevertheless, as ecological stability does not guarantee evolutionary stability, can such coexistence survive ceaseless mutation and selection? Why doesn't the producer in a cross-feeding pair adjust its strategy to retain all useful metabolites?

And can evolutionarily stable coexistence occur due to secreted metabolites even without mutually beneficial cross-feeding?

To explore the possibilities of evolutionarily stable coexistence when species create new nutrients, we used a simplified model to represent multi-step energy generation with a dual-role intermediate metabolite (Fig 5A). A single chemical energy source S is supplied into the chemostat. The pathway for processing S consists of four relevant reactions driven by designated enzymes: External S can be imported and converted into intermediate $I_{int}$ to generate ATP (with a corresponding fraction of the enzyme budget $\alpha_{ATP1}$).The intermediate has a dual role in energy production: on the one hand, it positively contributes to ATP production via a downstream reaction (with a fraction of the enzyme budget $\alpha_{ATP2}$); on the other hand, it negatively contributes to ATP production through product inhibition of the first energy-producing reaction. To deal with this negative effect of internal intermediate, cells may synthesize transporters (with a fraction of the enzyme budget $\alpha_{exp}$) to export intermediate out into environment, where it becomes external intermediate $I_{ext}$. By this reaction, cells can increase the dimension of chemical space from one (S) into two (S and $I_{ext}$). Cells can also import $I_{ext}$ into $I_{int}$ (with a fraction of the enzyme budget $\alpha_{imp}$), then use $I_{int}$ as an energy source via the second reaction. (See S1 Appendix for details.)

The metabolic strategy $\overrightarrow{\alpha}$ in this model has four components: $\overrightarrow{\alpha} = (\alpha_{ATP1}, \alpha_{ATP2}, \alpha_{exp}, \alpha_{imp})$. When we examine the maximizing strategies and maximal growth rates in the chemical space, three distinct classes of strategy emerge (Fig 5B). When S is abundant and $I_{ext}$ is low, the maximizing strategies have only two non-zero components, $\alpha_{ATP1}$ and $\alpha_{exp}$ (S7B Fig), meaning this class of species only imports S, for the first energy-generating reaction, then exports intermediate as waste. Therefore, we call strategies in this class "polluters" (blue section in Fig 5B, S7C Fig). When $I_{ext}$ is high while S is low, the maximizing strategies have two different non-zero components, $\alpha_{ATP2}$ and $\alpha_{imp}$ (S7B Fig), meaning this class of species relies solely on $I_{ext}$ as its energy source. We call these strategies "cleaners" as they clean up $I_{ext}$ from the environment, (green section in Fig 5B, S7C Fig). When there are comparable amounts of S and $I_{ext}$ present, a third class of maximizing strategies appears: these cells neither export nor import intermediates, but rather allocate all their enzyme budget to $\alpha_{ATP1}$ and $\alpha_{ATP2}$ to carry out both energy-producing reactions. We call species in this class "generalists" (red section in Fig 5B, S7C Fig).

As shown in Fig 5B, on the borders between classes of strategies in chemical space, the maximal growth contours turn discontinuously. These points of discontinuity, as in the example in the previous section, are chemical environments corresponding to evolutionarily stable coexistence of species from distinct metabolic classes. The classes of optimally coexisting species change with dilution rate. When the dilution rate is low ($d = 0.4$, Fig 5C), at the discontinuous point of the maximal growth contour, the corresponding two maximizing strategies are one polluter (species *Blue*) and one cleaner (species *Green*). Their supply lines span a gray region where both species *Blue* and species *Green* are required to create a steady-state environment on the maximal growth contour. As by assumption we are only supplying the system with S, the supply condition always lies on the *x*-axis of concentration space. For the supply condition shown by the black open circle in Fig 5C, polluter *Blue* creates a chemical environment (blue dot) far from the maximal growth contour. When the cleaner *Green* is added to the system, not only does the biomass of *Blue* increase (inset), but also the steady-state chemical environment moves to the discontinuous point of the maximal growth contour (cyan dot), where both *Blue* and *Green* occupy the peaks of their fitness landscapes (Fig 5D). This result is consistent with the long-term evolution experiment of *E. coli* and also intuitive: polluter *Blue* and cleaner *Green* form a mutually beneficial relationship by, respectively, providing nutrients and cleaning up waste for each other, thereby reaching an optimal cooperative coexistence.

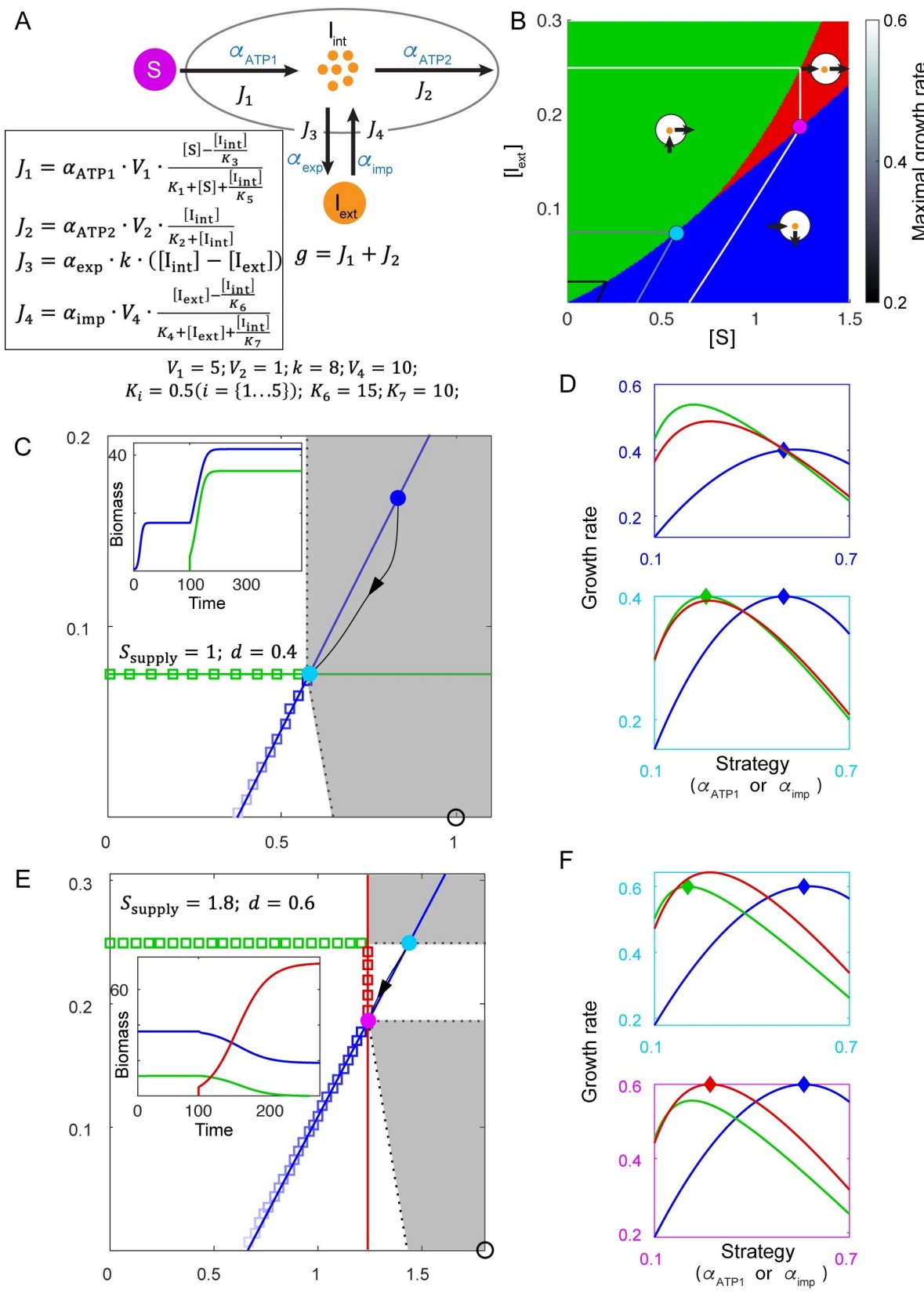

**Fig 5. Species creating new nutrient dimensions and achieving evolutionarily stable coexistence.** A. Metabolic model with a single supplied nutrient S. Cells allocate enzymes to convert S into internal intermediate $I_{int}$ and produce energy (denoted as "ATP"), export internal intermediate into the chemostat to become $I_{ext}$, import external intermediate, or consume $I_{int}$ to produce ATP. The growth rate is the sum of ATP production (see S1 Appendix). B. Three subclasses of maximizing metabolic strategies in chemical space are indicated by background color, and circles with arrows illustrate the metabolic strategies of each subclass. The maximal growth contours for three growth rates (0.2, 0.4, 0.6) are marked by black-to-white colors. C. At dilution rate $d = 0.4$, two maximizing strategies co-create a non-invasible environment. The maximal growth contour and the corresponding maximizing strategies are shown as colored squares. At a discontinuous point of the growth contour, the supply lines of two distinct metabolic strategies (*Green* and *Blue*) span a gray region, where any supply condition (e.g. black circle) requires two maximizing strategies to co-create the environment at the discontinuous point. The blue dot marks the environment created by species *Blue* alone, and the cyan dot marks the environment co-created by *Blue* and *Green*. Th black curve with arrows shows a trajectory in chemical space. Inset: time course of species biomass, with species *Green* added to the chemostat at time 100. D. The fitness landscapes for two environments in (C) indicated by corresponding colors of the boxes, reflecting the relationship between instantaneous growth rate and resource allocation strategy. For classes Blue and Red, the strategy is represented by $\alpha_{ATP1}$; for class Green the strategy is represented by $\alpha_{imp}$. E. Same as (C), except that the dilution rate is $d = 0.6$. Inset: time course of species biomass, starting with *Blue* and *Green*, with species *Red* added to the chemostat at time 100. F. Same as (D), except corresponding to the two steady-state environments shown in (E).

A quite different coexistence occurs at higher dilution rate ($d = 0.6$, Fig 5E). Growth contours at this dilution rate show two turning points, but neither are between the polluter and the cleaner class. One discontinuous point is between the cleaner class (green squares) and the generalist class (red squares), but the gray region spanned by the corresponding supply lines does not cover the *x*-axis and so does not represent an attainable coexistence when only S is supplied. The other discontinuous point is between the generalist class and the polluter class (blue squares). The gray region spanned by the supply lines of the corresponding two maximizing strategies of generalist class (species *Red*) and polluter class (species *Blue*) does cover the *x*-axis. Therefore, a supply condition with only S within the gray region (e.g., the black open circle) leads to the optimal coexistence of generalist *Red* and polluter *Blue* on the discontinuous point (purple dot), despite the fact that they do not directly benefit each other. Indeed, when the generalist *Red* is added to a system with polluter *Blue* and a cleaner *Green*, the cleaner *Green* goes extinct and the biomass of the polluter *Blue* decreases (inset). Nevertheless, the steady-state chemical environment is moved from a cyan dot lying inside the maximal growth contour to the purple dot lying on the maximal growth contour. In the environment of the cyan dot created by cleaner *Green* and polluter *Blue*, *Blue* is not on the top of the fitness landscape of the polluter class (Fig 5F, upper panel). By contrast, for the fitness landscape created by polluter *Blue* and generalist *Red* (Fig 5F, bottom panel), despite being lower in biomass, *Blue* occupies the top of the landscape. Therefore, the optimal coexistence of this polluter and this generalist does not arise from direct cooperation, but rather from collaborating to defeat other competitors.

## Discussion

Evaluating microbial metabolic strategies within an ecological context is the major focus of this work. Due to the intensity of competition in the microbial world, it is accepted that natural selection has extensively shaped microorganisms' internal resource allocation strategies and the regulatory mechanisms controlling these strategies [3, 4]. Therefore, a quantitative mapping from metabolic strategies to fitness consequences can further our understanding of both regulation and evolution [8]. Many previous studies of metabolic strategies directly assumed the optimization goal of microbial metabolism to be biomass gain, with the chemical environment acting as a fixed input [49–53], which simplifies the problem into a search for a maximum on a static "fitness landscape". However, in the natural world where metabolic strategies compete and evolve, the feedback between species and their environment produces an intrinsically dynamic fitness landscape in which the actions of one species can influence the fitness of

all species [54]. One profound example is the Great Oxygenation Event, when cyanobacteria created an oxygen-rich atmosphere [55], causing a massive extinction of anaerobic bacteria but also stimulating an explosion of biodiversity [56]. Therefore, metabolic strategies need to be assessed within an ecological context, taking into consideration not only how species respond to the environment, but also how species construct their own environment.

For the past thirty years, researchers have been utilizing various mathematical tools to study flexible fitness landscapes that change with space, time, and population composition. The prerequisite for an intrinsically dynamical fitness landscape–that the species composition influences the fitness of all species in the system–takes a particularly simple form in chemostat-type resource-competition models: Microbes shape their local environment by exchanging metabolites within a shared chemical environment, which determines the growth rate of all cells. Focusing on the chemical environment created by microbial metabolism, we exhibited a set of intuitive and general procedures for analyzing strategies within various metabolic models. Namely, we showed that to compare a set of fixed strategies, the geometric relationships between their growth contours and their steady-state chemical environments yield an immediate prediction for the outcome of invasion. In searching for optimality over the continuous family of strategies, the "maximal growth contour" envelope of all growth contours provides candidates, and the supply condition selects the non-invasible strategy or strategies from among these candidates via the flux-balance curve. Such selection of optimal strategies also supports the conclusion that having the fastest growth rate in an environment does not necessarily imply being the most competitive strategy, as this strategy may shift the environment in an unfavorable direction. To be non-invasible, strategies also need to be able to construct the environment for which they are best suited. Finally, evolutionarily stable coexistence occurs at the discontinuous points of the maximal growth contour.

The deformability of the fitness landscape also has implications for microbial community assembly, particularly in establishing the criteria for coexistence on both ecological and evolutionary timescales. Given resource allocation trade-offs, the growth contours of any pair of strategies must intersect, clearly demonstrating why trade-offs prevent a single species from unconditional dominance, allowing various forms of intransitivity under different metabolic models. For example, the crossings of different growth contours take an extreme form in the substitutable-nutrient model with exact trade-offs, where all strategies intersect at the same chemical environment, enabling a flat fitness landscape for unlimited coexistence [27]. In subsequent works that assume non-exact trade-offs and spatial structure [42], or temporal variation in nutrient supply with immigration [57], a large number of species still coexist, demonstrating how a nearly-flat fitness landscape can promote diversity. Moreover, we also demonstrated that ecological or evolutionary adaptation towards a higher growth rate promotes the emergence of keystone species, which ultimately create a flat landscape for all species. In a recent publication [58], species are allowed to adjust their metabolic strategies over time to maximize their relative fitness. These adaptive strategies yield similar results to our evolutionary adaptation via invasion: even when the supply condition initially does not favor coexistence, the adaptation of metabolic strategies self-organizes the population into a state of coexistence. Our results suggest that even though mutations which increase the total enzyme budget of species disrupt coexistence [59], mutations that adjust metabolic strategies can promote biodiversity.

The essential-nutrient metabolic model exhibits a different form of fitness landscape: the ability of each species to create an environment that favors itself promotes multistability and spatial coexistence (S6 Fig). Multistability between species adopting discrete nutrient utilization strategies [26, 60] has been intensively investigated in recent years. Here, we further demonstrate that when species are allowed to evolve in a continuous strategy space, a large number

of strategies that cannot invade each other emerge as candidates for multistability. At the end of a long chain of invasions, a non-invasible strategy finally appears, demonstrating the criteria for "evolutionarily stable" strategies.

On the evolutionary time scale where mutation/adaptation allows searches for the "most suitable" strategies among infinite possibilities, an ongoing threat to diversity is that selection may produce a supreme winner that takes over the habitat. With the dynamic fitness landscape and the maximal growth contour approach, we showed that the condition for evolutionarily stable coexistence is indeed restricted, occurring only at the discontinuous points of the maximal growth contour. Nevertheless, via the species-environment feedback, a large number of supply conditions can self-organize to these discontinuous points, where multiple species co-create a non-invasible environment where they jointly locate on the peak of the fitness landscape. Among the several types of evolutionarily stable coexistence we investigated, the most unexpected is between a "polluter" and a "generalist", demonstrating that metabolite secretion can lead to diversity even in the absence of cross-feeding. This observation complements and extends the previous consensus that coexistence can be enabled by unilateral or mutually beneficial cross-feeding [23–25, 30]. We also observed that species that compete rather than cooperate with each other can achieve evolutionarily stable coexistence by jointly creating the "worst possible" environment for each other's competitors. To our knowledge, this kind of non-cooperative evolutionarily stable coexistence has not previously been investigated in detail.

Many future directions can follow this work. From the perspective of experiment, our framework can assist in analyzing and interpreting results of microbial evolution in the lab [61], where the continual emergence of new mutants under defined experimental conditions suggests an intrinsically dynamic fitness landscape. From the perspective of theory, we do not yet have a rigorous mathematical theorem concerning the conditions for discontinuity of the maximal growth contour, nor proof that discontinuity necessarily leads to evolutionarily stable coexistence. Theoretical developments paralleling those on the general existence of ecologically stable states [15, 62] would bring a more comprehensive understanding of evolutionarily optimal states in metabolic models. Besides, the metabolic models considered in this work are highly simplified. Going forward, more detailed and experimentally-based models can be examined using the same graphical and dynamical fitness landscape framework.

## Methods

Programs for this work are coded in MATLAB R2018a. Please see S1 Appendix for the details of equations, parameters, and analytical solutions.

## Supporting information

**S1 Appendix. Model description.**
(DOCX)

**S1 Fig. How supply concentrations and dilution rate separately influence the shapes of nullclines and the steady-state environment.** A. Various supply concentrations can lead to the same steady-state chemical environment. Background color indicates the growth rate of cells as a function of nutrient concentrations $c_a$ and $c_b$, with the growth contour shown by the red curve. The supply line for the steady-state environment (purple dot) is shown as a dotted black line. Different supply concentrations ($c_{a,\text{supply}}$ and $c_{b,\text{supply}}$) along the supply line are marked by purple, cyan, and blue circles, with the corresponding flux-balance curves shown in the same colors. B. Dilution rate can flip nutrient limitation. The external supply condition is

marked by a blue circle, and the flux-balance curve for this supply is shown in the same color. Three growth contours with increasing dilution rates are shown from yellow to deep red, and the corresponding steady-state environments are shown as colored dots.
(TIF)

**S2 Fig. Nutrient supply shifts the relationship between RNA/Protein ratio and growth rate in chemostat.** A. The relationship between ribosome abundance represented by RNA/Protein ratio (*y*-axis) and growth rate (*x*-axis) of *E. coli* cultured in chemostats from phosphorus limitation (P-limited, green open circles and dotted line) to nitrogen limitation (N-limited, blue open circles and dotted line). Starting from the P-limited condition, data for decreasing the supply concentration of nitrogen by 2, 5, and 10-fold are shown as solid dots and corresponding best-fit lines. Each measurement was repeated three times and standard errors are shown by bars. C. Same as (B), but for phosphorus and carbon limitation instead of phosphorus and nitrogen limitation. Starting from the P-limited condition, data for decreasing the supply concentration of carbon by 2, 5, and 10-fold are shown as solid dots and corresponding best-fit lines.
(TIF)

**S3 Fig. The substitutable-nutrient metabolic model in 10-dimensional chemical space.** A. The enzyme allocation strategies of the initial species (Init) and the 9 opportunist species (Opp 1–9) that appear in succession in Fig 2E. B. The instantaneous growth rates of $10^4$ randomly generated enzyme allocation strategies, under the steady-state chemical environments created by consecutively adding the species shown in (A) into the existing consortia. C. Example of the "outermost" surface formed by multiple growth contours, as highlighted in yellow. D. Rescaled radar plot for the "most favorable environment" for each strategy that appears in (A). Dots represent the centroids of polygons representing the chemical environments created after adding each species (see S1 Appendix "*Metabolic model with substitutable nutrients*" for details).
(TIF)

**S4 Fig. Stochastic behaviors in the rock-paper-scissors fitness landscape A.** Example of a metabolic model with a trade-off in allocation of internal resources for import and assimilation of three substitutable nutrients, with all three nutrients contributing additively to growth. Species 1 (*Red*), species 2 (*Blue*), and species 3 (*Green*) allocate resources differently (see S1 Appendix). B. The fitness of Species 1, 2, and 3 in the steady-state environment constructed by species 1, 2, and 3. C. Growth contours (surfaces), flux-balance curves (lines), and steady-state nutrient concentrations (dots) for the three species in a three-dimensional chemical space. Black curves with arrows show the system's limit-cycle trajectory. D. The upper panel shows the time course of species biomass in the chemostat for the limit cycle in (C). The bottom panel shows how the fitness landscape changes with time over one period of the oscillation. E. Stochastic simulation of the model shown in Fig 3, where species are never allowed to drop to zero biomass. F. Same as (E), except that species are considered to become extinct after dropping to zero biomass.
(TIF)

**S5 Fig. The chain of invasion in 10-dimensional chemical space.** A. Population dynamics of the metabolic model with 10 essential nutrients, where each newly introduced species has the fastest-growing strategy in the steady-state environment created by the existing consortium. B. The enzyme allocation strategies of the species that appear in succession in (A). C. The instantaneous growth rates of the strategies that appear in (A) under the steady-state environments created by the existing consortia. Gray arrows indicate addition of the new species with the fastest-growing strategy, and black arrows indicate the change of the steady-state chemical

environment induced by adding this new species. Black contours indicate growth rate equal to dilution rate. D. The chain of invasion. Each colored dot represents one species in (A). An arrow from species $i$ to species $j$ indicates successful invasion of $j$ into the environment created by $i$ alone. E. Rescaled radar plot for the "most favorable environment" for each strategy that appears in (A). Dots represent the center of the chemical environment created by each species. (TIF)

**S6 Fig. Steady-state spatial heterogeneity for linked chemostats.** With initial seeding of two species, one at each of the two ends of a chain of chemostats, a steady-state gradient of species biomass density spontaneously emerges accompanied by a gradient of nutrient concentrations, even though the supply conditions and dilution rates are identical for all the chemostats. A. Schematic of $k_{tot}$ linked chemostats exchanging medium and cells via leakage, described by Eqs. S46-S47. The two species in the chemostats (*Blue* and *Red*) are the same bistable pair as in Fig 3B and the leakage rate is $l = 1$. B. The species composition along 20 linked chemostats for the system in (A). Species colors correspond to those in Fig 3B, with species *Blue* having $\alpha_a =$ 0.35 and species *Red* having $\alpha_a = 0.65$. The dashed black curve shows the sum of the two biomass densities. The initial condition was cell-free chemostats with a small amount of *Blue* added to Chemostat 1 and small amount of *Red* added to Chemostat 20. C. Concentrations along the 20 chemostats for nutrient $a$ (green) and nutrient $b$ (cyan) for system in (A). D. The fitness landscape along the chain of chemostats. The $x$-axis is the 20 linked chemostats, and the $y$-axis is the metabolic strategy represented by $\alpha_a$. Color indicates the growth rate of species adopting the given strategy in the $k$-th chemostat. (TIF)

**S7 Fig. Maximizing strategies in chemical space.** A. For each environment in the chemical space, the maximizing resource allocation strategies that maximize growth rates for the model in Fig 4A. Each strategy is represented by the four elements $[\alpha_a, \alpha_b, \alpha_{ab}, \alpha_{ba}]$, and values for each element are shown by a heatmap. Black-to-white curves are the maximal growth contours for $d = 0.1, 0.2, 0.3, 0.4$. B. For each environment in the chemical space, the maximizing resource allocation strategies that maximize growth rates for the model in Fig 5A. Each strategy is represented by the four elements $[\alpha_{ATP1}, \alpha_{ATP2}, \alpha_{exp}, \alpha_{imp}]$, and values for each element are shown by a heatmap. Black-to-white curves are the maximal growth contours for $d = 0.2$, 0.4, 0.6. C. Schematic representations of the three classes of maximizing strategies appearing in (B). (TIF)

## Acknowledgments

We thank Simon Levin for insightful discussions.

## Author Contributions

**Conceptualization:** Zhiyuan Li, Zemer Gitai, Ned S. Wingreen.

**Data curation:** Zhiyuan Li, Bo Liu, Sophia Hsin-Jung Li, Christopher G. King, Zemer Gitai.

**Formal analysis:** Zhiyuan Li, Bo Liu.

**Funding acquisition:** Ned S. Wingreen.

**Methodology:** Zhiyuan Li.

**Supervision:** Ned S. Wingreen.

**Visualization:** Zhiyuan Li.

**Writing – original draft:** Zhiyuan Li.

**Writing – review & editing:** Zhiyuan Li, Ned S. Wingreen.

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
