## [Decision Letter · Decision Letter 0]

14 Feb 2020

Dear Prof. Wingreen,

Thank you very much for submitting your manuscript "Modeling microbial metabolic trade-offs in a chemostat" for consideration at PLOS Computational Biology.

As with all papers reviewed by the journal, your manuscript was reviewed by members of the editorial board and by several independent reviewers. In light of the reviews (below this email), we would like to invite the resubmission of a significantly-revised version that takes into account the reviewers' comments.

We cannot make any decision about publication until we have seen the revised manuscript and your response to the reviewers' comments. Your revised manuscript is also likely to be sent to reviewers for further evaluation.

Sincerely,

Jacopo Grilli

Associate Editor

PLOS Computational Biology

Stefano Allesina

Deputy Editor

PLOS Computational Biology

Reviewer's Responses to Questions

**Comments to the Authors:**

Reviewer #1: My review report is attached as a PDF file.

Reviewer #2: The authors use an explicit model of resource competition to search for optimal resource allocation strategies in a continuous growth situation, e.g. in a chemostat. They use a general framework—which they have introduced before—that assumes metabolic trade-offs in the form of a linear allocation of limited resources in different metabolic pathways.

The authors start by framing a model and a corresponding graphic visualization to interpret the outcome. They then proceed with a series of examples, with relatively simple and intuitive assumptions, to show different types of complex dynamics that can emerge. The focus throughout the paper is on how the trade-off in allocating resources can lead to ecological dynamics.

The manuscript does a great job of putting its contribution in the context of the existing work. It has a logical flow and is accessible for both general and expert readers, in my opinion (although defining some of the terms in the introduction could help making the paper more readable for non-experts). I have no major concerns regarding the paper (minor suggestions are listed below).

Minor comments:

1. I am not sure if I fully understand what happens in Fig 2. Is the flat landscape a result of a particular choice of parameters for species Blue and Red? Additionally, doesn’t the presence of other species reshape the landscape? Or does the landscape become flat with any combination that has both species with small alpha_a and species with large alpha_a?

2. I think it would be helpful to include the list of relevant model parameters and their values in each figure as a point of reference.

3. Although conceptually generalizable, the visualization approach works well only with one or two environmental resource (with three resources, as in Fig 3, the visual representation is not nearly as clear). I admit that even with two resources the authors have shown a very rich range of possibilities, but it would be helpful to include your suggestions about how to scale up to more resources.

Reviewer #3: The manuscript presents a very broad exploration of consumer-resource models in constantly diluted chemostats with general functions describing species’ growth rates, nutrient consumption and secretion rates, and their (potentially variable and evolvable) metabolic strategy alpha. The paper adopts the geometric view of Tilman 1982 representing ecosystem dynamics in a multidimensional space of nutrient concentrations inside the chemostat. The scope of the paper is so broad that sometimes it is difficult to separate truly novel results from well-established facts in consumer-resource modeling literature. However, I believe that with a proper narrowing of the scope and streamlining of explanations the paper would be publishable in PLoS Comp Biology. Here are my comments, questions, and recommendations roughly in the order the models were presented in the manuscript:

1) Lines 305-306. Invoking the model as an explanation for the paradox of the plankton is rather weak and should be removed. Indeed, a long-term coexistence of species requires near miraculous coincidence: all species have to have exactly the same proteome budget and differ only by their allocation strategies alpha (sum alpha =1). While this is plausible for subpopulations within one genetically identical strain, it is rather unlikely even for closely related strains of the same species (see Lenski experiments and recent studies by Ben Good and Oskar Hallatschek about two types of mutations: one type changing alpha and another type changing the overall growth rate). Given that in the paradox of the plankton a very diverse set of organisms coexists, it is unlikely to be explained by these arguments. In fact, this part of the study is purely pedagogical and does not describe any new results.

2) Lines 336-337. The model described in this section is potentially new and interesting. However, I did not understand from the main text how exactly a mismatch between allocation towards nutrient import and conversion can lead to rock-paper-scissors oscillations. If it is described in SI, it should be moved to the main text. How is this model different from Huisman et al 1999, especially from a more detailed paper by the same authors: Jef Huisman and Franz J Weissing, ECOLOGY 2001 “Biological conditions for oscillations and chaos generated by multispecies competition”?

3) The part of the manuscript describing multistability needs to be put in context of a recent 2019 paper from Maslov lab dedicated exclusively to this topic. What are authors’ new insights into the topic of multistability? Also, it has not been spelled very clearly in this manuscript that in MacArthur type models multistability is possible only when different species have different yields on the same nutrient. This clear and simple message has been lost in the general approach adopted in this manuscript.

4) Evolutionary stable strategies resisting invasion by other species with different metabolic strategies is a truly novel ingredient of this manuscript (see though a recent preprint from Martian and Suweis labs with similar discussion). However, at this point in the manuscript I was so exhausted plowing through previous results that I did not understand the main point authors are trying to make here. I would like authors to sharpen their message in this section. They should also put their results in context of Tikhonov and Monasson 2018 J Stat Phys paper on evolution of parameters in consumer-resource models.

5) The discussion of creating metabolic niches by secreting metabolic byproducts is not new. What is missing is the discussion of authors insights in context of Marsland et al “Available energy fluxes drive a transition in the diversity, stability, and functional structure of microbial communities” and Goyal et al PRL 2018.

I would be happy to review a more streamlined version of this manuscript again.

**Have all data underlying the figures and results presented in the manuscript been provided?**

Reviewer #1: Yes

Reviewer #2: Yes

Reviewer #3: Yes

PLOS authors have the option to publish the peer review history of their article (what does this mean?). If published, this will include your full peer review and any attached files.

Reviewer #1: No

Reviewer #2: No

Reviewer #3: No
---

## [Decision Letter · Decision Letter 1]

16 Jul 2020

Dear Prof. Wingreen,

We are pleased to inform you that your manuscript 'Modeling microbial metabolic trade-offs in a chemostat' has been provisionally accepted for publication in PLOS Computational Biology.

Best regards,

Jacopo Grilli

Associate Editor

PLOS Computational Biology

Stefano Allesina

Deputy Editor

PLOS Computational Biology

Reviewer's Responses to Questions

**Comments to the Authors:**

Reviewer #1: I am personally satisfied by the authors' replies.

I would just like to point out that at some points in the revised manuscript (lines 214, 255, 343, 405, 409 and 418) there are some references to equations in the text that have not been compiled correctly, and as a result no equation number is shown.

Reviewer #2: In my opinion, the revisions have adequately addressed the concerns raised by referees. The work is now more clearly placed within the context of previous reports and the figures are easier to follow.

Minor typo: some references to equation numbers are missing (e.g. lines 214, 255-256, 393, 405, 409, and 418).

**Have all data underlying the figures and results presented in the manuscript been provided?**

Reviewer #1: Yes

Reviewer #2: Yes

PLOS authors have the option to publish the peer review history of their article (what does this mean?). If published, this will include your full peer review and any attached files.

Reviewer #1: No

Reviewer #2: No

---

## [Editor Report · Acceptance letter]

20 Aug 2020

PCOMPBIOL-D-19-02227R1 

Modeling microbial metabolic trade-offs in a chemostat

Dear Dr Wingreen,

I am pleased to inform you that your manuscript has been formally accepted for publication in PLOS Computational Biology. Your manuscript is now with our production department and you will be notified of the publication date in due course.

With kind regards,

Matt Lyles
